# Application of Image Processing and 3D Printing Technique to Development of Computer Tomography System for Automatic Segmentation and Quantitative Analysis of Pulmonary Bronchus

**Chung Feng Jeffrey Kuo** [1] **, Zheng-Xun Yang** [1] **, Wen-Sen Lai** [2,3,†] **and Shao-Cheng Liu** [3,*,†]

1  Department of Materials Science and Engineering, National Taiwan University of Science and Technology, Taipei 106, Taiwan
2  Department of Otolaryngology-Head and Neck Surgery, Taichung Armed Forces General Hospital, Taichung 411, Taiwan
3  Department of Otolaryngology-Head and Neck Surgery, Tri-Service General Hospital, National Defense Medical Center, Taipei 114, Taiwan
*  Correspondence: m871435@ndmctsgh.edu.tw; Tel.: +886-2-8792-7192; Fax: +886-2-8792-7193
†  Co-correspondence author.

**Abstract:** This study deals with the development of a computer tomography (CT) system for automatic segmentation and quantitative analysis of the pulmonary bronchus. It includes three parts. Part I employed an adaptive median and four neighbors low pass filters to eliminate the noise of CT. Then, k-means clustering was used to segment the lung region in the CT data. In Part II, the pulmonary airway was segmented. The three-grade segmentation was employed to divide all pixels in the lung region into three uncertain grades, including air, blood vessels, and tissues, and uncertain portions. The airway wall was reformed using a border pixel weight mask. Afterwards, the seed was calculated automatically with the front-end image masking the aggregation position of the lung region as the input of the region growing to obtain the initial airway. Afterwards, the micro bronchi with different radii were detected using morphological grayscale reconstruction to modify the initial airway. Part III adopted skeletonization to simplify the pulmonary airway, keeping the length and extension direction information. The information was recorded in a linked list with the world coordinates based on the patients' carina, defined by the directions of the carina to the top end of the trachea and right and left main bronchi. The whole set of bronchi was recognized by matching the target bronchus direction and world coordinates using hierarchical classification. The proposed system could detect the location of the pulmonary airway and detect 11 generations' bronchi with a bronchus recognition capability of 98.33%. Meanwhile, 20 airway parameters' measurement and 3D printing verification have been processed. The diameter, length, volume, angle, and cross-sectional area of the main trachea and the right and left bronchi, the cross-sectional area of the junction, the left bronchus length, and the right bronchus length have been calculated for clinical practice guidelines. The system proposed in this study simultaneously maintained the advantages of automation and high accuracy and contributed to clinical diagnosis.

**Keywords:** bronchus; image processing; computer-aided detection; k-means; 3D reconstruction

**MSC:** 92-04; 68T07; 68T37; 68U10; 68W99

## 1. Introduction

Lung cancer is the most familiar cancer, with its proportion among all cancer types is as high as 11.6% [1]. With careful evaluation of lung lesions, such as non-small-cell lung cancer (NSCLC) and pulmonary fibrosis, the pulmonary lobectomy is an effective and direct therapy [2,3]. Evaluation and preoperative planning are necessarily based on the

computer tomography (CT) data [4–6]. This study developed an objective and accurate system from image processing techniques to analyze the lung structure and provide the position information for clinicians based on the intrapulmonary bronchi. The CT data reading time was shortened for clinicians. Additionally, the information for judging and evaluating the effects of an operation was increased.

The literature review for the system was discussed in three steps, including lung segmentation, pulmonary airway detection, and bronchial recognition. The lung segmentation can accelerate subsequent image processing and assist in finding out relevant information for pulmonary airway detection. Therefore, lung image segmentation is very important. Pulmonary airway detection is a vital part of finding the target. The pulmonary airway is an indispensable part of pulmonary function. Clarifying and analyzing the pulmonary airway is significantly helpful to doctors' preoperative planning. Bronchial recognition is the most important part of a computer-aided system. Detecting the bronchus position by analyzing the pulmonary airway can enhance the construction of overall lungs for the computer-aided system.

### 1.1. Lung Segmentation

The left and right lungs are subdivided into five pulmonary lobes, including the upper left pulmonary lobe and lower left pulmonary lobe on the left, the upper right pulmonary lobe, middle right pulmonary lobe, and lower right pulmonary lobe on the right. These lobes are separated by fissures. Van Rikxoort et al. [7] applied region growing and morphological smoothing to segment lung fields. The scans that are likely to contain errors in some abnormal cases are segmented by multiatlas segmentation. By using a 3D region grown method, De Nunzio et al. [8] achieved human airway (trachea and bronchi) segmentation with suitable stop conditions and wavefront simulation. Accurate identification of all the pulmonary nodules can be ensured by the 3D morphology operations. Diciotti et al. [9] presented the user interaction process to allow for the introduction of the expert's knowledge in a simple and reproducible manner. Adopting the geodesic distance in a multithreshold image representation allows the definition of a segregation process based on gray-level similarity and object shape. Pu et al. [10] presented a shape analysis strategy termed "break-and-repair" to facilitate automated medical image segmentation. The principal curvature analysis can be used to identify and remove problematic areas. Implicit surface fitting of radial basis function (RBF) can be used to achieve closed (or complete) surface boundaries.

Prabukumar et al. [11] used fuzzy c-means (FCM) and region growing segmentation algorithms for the nodule of interest from the CT lung images. The SVM classifier was trained using extracted features to classify lung cancer. Xu et al. [12] proposed lung parenchyma segmentation in CT images using a CNN trained with a dataset generated with a clustering algorithm. The designed CNN architecture consists of one convolutional layer with six kernels, one max pooling layer, and two fully connected layers. Helen et al. [13] developed an improved 2D Otsu algorithm [14] based on particle swarm optimization (PSO) to reduce the complex computation and computation time. The PSO was used to find the optimal threshold for the segmentation and extraction of lung parenchyma in less time. Ahmad et al. [15] showed the segmentation technique for computed tomography images to segmented liver based on deep learning. The stack autoencoder is used to learn features from the images. Qadri et al. [16] used autoencoder-based patch classification to segment vertebrae. The extracted features were fed into a logistic regression classifier to fine-tune the model, and a sigmoid classifier was used to discriminate vertebrae and non-vertebrae patches.

The conventional lung segmentation methods include thresholding, region growing, and clustering. Direct thresholding requires a large quantity of predetermined data for calculating appropriate thresholds. It requires varying degrees of methods to repair the result of the preliminary threshold. The region growing is an iterative method, and it cannot predict the occurrence of diffusion. The clustering method is a preferable lung segmentation

method [12]. The classification result of k-means is used to train the CNN. A large quantity of predetermined data is required, but the result can be calculated faster. This study improved the method. The k-means formed of all data were combined with connected morphological representation and the hole-filling method. The system rapidly extracted the peripheral contour block of the lungs. It got rid of the restriction of predetermined data and recorded the centroid of k-means for future use.

### 1.2. Pulmonary Airway Detection

Tschirren et al. [17] developed an airway segmentation method based on fuzzy connectivity for lung airway detection. Small adaptive regions of interest followed the airway branches as they were segmented. Bauer et al. [18] proposed a graphics-based framework for reconstruction of the airway tree from CT scan images. Potential airway branches and candidate connection sites can be efficiently identified and represented by a graph structure with weighted nodes and edges. A subset of airway branches and connections was selected based on graph weights derived from image features, and an optimized algorithm can, thus, generate airway detection results. Selvan et al. [19] introduced graph refinement-based airway extraction using mean-field and graph neural networks (GNNs). The mean-field approximation (MFA) was applied to approximate the posterior density over the subgraphs from which the optimal subgraph of interest could be estimated. Mean-field networks (MFNs) were used for inference based on the interpretation that iterations of MFA could be seen as feed-forward operations in a neural network. By using GNNs, the supervised learning approach can be seen as a generalization of MFNs. For accurate airway lumen segmentation, fuzzy connectedness theory was employed [20] to spatially constrain the Markov random walk. Fetita et al. [21] developed a generic and automated airway segmentation approach to deal with a large spectrum of multi-slice computed tomography (MSCT) protocols by exploiting a combined morphological aggregative methodology. Charbonnier et al. [22] emphasized and refined airway segmentation by using leak detection, a classification problem in which convolutional neural networks were trained for classification. Rosell et al. [23] presented a three-stage segmentation method for the 3D reconstruction of the tracheobronchial tree from CT scans. Using adaptive region growing, they proposed gross segmentatio, which takes reconstruction of the main airway tree as a first step. Next, any potential airway regions were identified to enable finer segmentation by using local information based on a 2D process that enhanced the bronchial wall. The final step was to connect any isolated bronchi to the main airway using morphological reconstruction procedures and path planning techniques. Fabijańska et al. [24] used a two-pass region growing algorithm for segmenting airway trees from multidetector computed tomography (MDCT) chest scans. The first pass was applied to obtain the initial (rough) airway tree. The second pass aimed at refining the tree based on the morphological gradient. Such a mechanism prevents leakages into the lungs and avoids falsely detected branches. Aykac et al. [25] used grayscale morphological reconstruction to identify candidate airways on CT slices and then reconstructed a connected 3D airway tree. After segmentation, airway branch points were estimated from the connectivity changes in the reconstructed tree.

The pulmonary airway segmentation methods in the literature can be divided into three classes, as follows: the iteration method achieved by iterations in a specific condition, the training method guided by predetermined data, and the correction method formed of k rough results and k + 1 fine results. The study was completed by a single method of mixing. The iteration method took a specific condition as the endpoint, but the iteration sometimes did not converge. The training method obtained ground truth by manually labeling data. It had a longer training time than the iteration method, but it can be called rapidly after training. The correction method looked for possible voxels based on the characteristics in the dataset to create a rough result. Based on the rough result, detailed results were searched in the data. It is the most intuitial method free from predetermined data.

### 1.3. Bronchial Identification

Mori et al. [26] performed anatomical labeling, and the names of the branches are automatically presented and shown in the currently rendered image in real-time. Based on the anatomical relationships and statistical intensity distribution among different organ and tissue regions, Zhou et al. [27] divided the CT image into target organ and tissue regions, sequentially. The basic rules of the processing flow include region extraction, detail correction, and structure recognition. The principal parameters of each process are automatically and dynamically self-optimized to adapt to different patient cases. By combining information from fissures, bronchi, and pulmonary vessels, Lassen et al. [28] performed a marker-based watershed transformation on CT scans to subdivide the lungs into lobes. By analyzing auto-labeled bronchial trees, the lobar markers are calculated and data is integrated to achieve fine segmentation, even in incomplete fissures. Tschirren et al. [29] performed both matchings of branch points and anatomical labeling of in vivo trees without human intervention and within a short computing time. No hand-pruning of false branches is required. Feragen et al. [30] presented a new atlas-based algorithm for anatomical branch labeling of airway trees based on the geodesic tree-space distances between them. Using tree-space distances, the algorithm evaluates how well the proposed branch labeling matches the labeled airway tree training set and determines the optimal labeling. Nadeem et al. [31] used hierarchical branch-level features from the current, ancestral, and descendant branches. The first step distinguishes candidate anatomical branches from insignificant topological branches. The second step is to perform lobe-based classification of the anatomical labels of valid candidate branches.

Misunderstandings and topological variation in pulmonary airway detections are the challenges in bronchial identification. Using skeletonization to extract the trend information of the pulmonary airway is a generally accepted bronchial identification procedure. The selection of the skeletonization system and the partial leak in pulmonary airway detection may cause misrecognition. This study used hierarchical classification [30] to identify anatomic Grade I, Grade II, and Grade III. After identification, all subnodes subordinate to these nodes were defined as the same class, and the airway masks were classified accordingly. The masks were reclassified as masks named by different bronchi.

The main contributions of the article can be summarized as follows:

1. The research problem has been clearly defined and the motivation has been clearly presented according to the development of a computer tomography system for the segmentation and quantitative analysis of pulmonary bronchus;
2. The literature review was discussed, including lung segmentation, pulmonary airway detection, and bronchial recognition;
3. This article has been based on the technical bottlenecks to be overcome and the corresponding key technologies to be developed;
4. The aims of the manuscript, which are to develop an objective and accurate system from image processing techniques successfully to analyze the lung structure and provide the position information for clinicians based on the intrapulmonary bronchi, will be shown. Meanwhile, 3D printing verification for the measurement of airway parameters have been processed.

## 2. Methods

### 2.1. Data Preprocessing

The image processing technique was used in the lung CT. Data preprocessing, lung circling, pulmonary airway circling, data systematization, and bronchial identification provided an intrapulmonary bronchi-based positioning system. Figure 1 is the processing flow chart of this system.

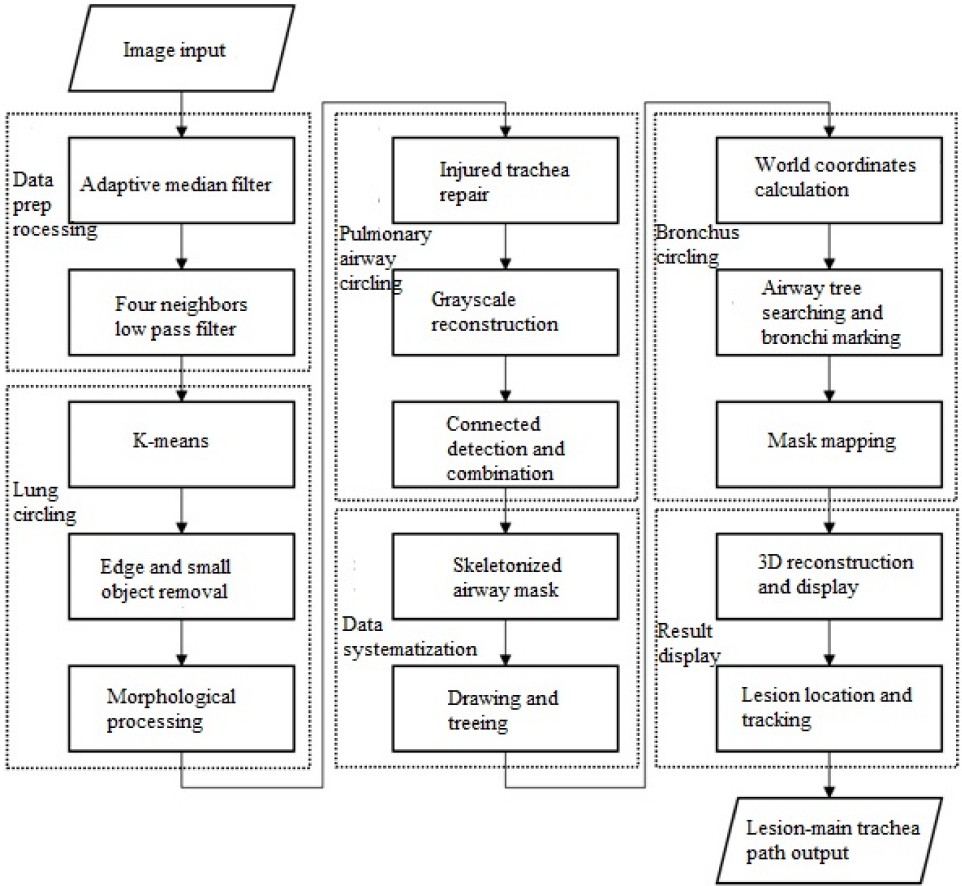

**Figure 1.** Image processing procedure.

### 2.1.1. Adaptive Median Filter

The adaptive median filter is used when the noise ratio is high (proportion of noise was larger than 0.2), or the high-frequency signal (e.g., tracheal wall and airway) is the main target to maintain the fine texture in the medical image. The variance in the target mask and the amount of variation of maximum and average values were calculated to maintain the detailed texture while filtering the noise. The pixel in an arbitrary position $(i, j)$ in an image $O$, and the $m \times n$ region of $I_k(i, j)$ was masked, as expressed in Equations (1)–(4):

$$\alpha_k = Max(I_k(i, j)) \tag{1}$$

$$\sigma_k = \sqrt{\frac{\sum_{i=1}^{N_{num}} (I_k(i, j) - \overline{I}(i, j))^2}{N_{num}}} \tag{2}$$

$$\gamma_k = \frac{\alpha_k}{\sigma_k} \tag{3}$$

$$M(i_{\text{centroid}}, j_{\text{centroid}}) = \begin{cases} \sum_{x=0}^{n} \sum_{y=0}^{m} Med(I(i, j)), \ when \ \gamma_k > T \\ O(i, j), \ when \ \gamma_k < T \end{cases} \tag{4}$$

where $\alpha_k$ is the maximum value, representing the maximum value in the mask $I_k(i, j)$, $\sigma_k$ represents the variance in the mask $I_k(i, j)$, $\overline{I}(i, j)$ is the average value in the mask, $N_{num}$ is the number of elements in the mask, $\gamma_k$ is the separation index for judging the internal energy of the image in $O(i, j)$, $M$ is the image output after filtering, and $T$ is the adjustable parameter for controlling the sensitivity of the adaptive median filter.

### 2.1.2. Four Neighbors Low Pass Filter

The four neighbors low pass filter is the minimum balanced mask using "four neighbors" when the objective mask [32] is selected. The feature selected by the objective mask is mixed with additional characteristics. This study needed to enhance the injured airway wall's gray level and repair the damaged pixel in the subsequent procedure. The structure could be mixed with the peripheral features at the shortest Euclidean distance (nearest spatial character), and the image could be smoothed with minimal image feature damage.

### 2.2. Lung Circling

The implementation challenge is the clustering method chosen to segment the lungs, the gas, gas-like, and tissue parts in the image. This study used k-means [33] to allocate $n$ points in a group of data to k groups of classes and to minimize the Euclidean distance from the points of all classes to the class center (cluster center).

After the preliminary segmentation using k-means, the objects were distinguished by representation, and the non-target objects were removed. The labels with boundary pixels and those that were not the maximum volume were removed. In the lung circling of this study, the blood vessels in the lungs might generate holes in the circled object, which were processed by morphology. The hole points of the object in the original image were dilated continuously and united with the complementary set of target objects until the object did not change after the dilation operation, as expressed in the following Equation (5):

$$X_k = (X_{k-1} \oplus se) \cap I^c, k = 1, 2, 3 \ldots \tag{5}$$

where $X_k$ is the final result of iteration, $\oplus$ is the dilation operator, $se$ is the structuring element, and $I^c$ is the complementary set of the original image.

### 2.3. Pulmonary Airway Circling

The implementation challenge is due to the restrictions of CT, the effects of diseases and noise, and the fact that the pulmonary airway wall is sometimes ruptured or injured. To identify a pixel in the gray image as the foreground (region of interest) or background (noise), it should be repaired before the pulmonary airway is circled [23]. To determine the airway or blood vessel in the target pixel, the values of pixels in six directions within 100 grids around a pixel are observed by identifying the brightness as foreground or background.

Grade 2 is when the pixel is brighter than peripheral pixels (foreground).

Grade 1 is when the pixel is not darker or brighter than peripheral pixels (uncertain pixel).

Grade 0 is when the pixel is darker than peripheral pixels (background).

To find the criteria, the pixel sets at 6 angles within 100 grids are required, and the angles are defined by the following Equation (6):

$$a_i = 30i + b, i = [1, 2, 3, 4, 56], b = rand([0, 30]) \tag{6}$$

where $a_i$ is the viewing angle at one of six times. Let $P_i(x, y)$ be the pixel set at an angle $a_i$. The criterion Equation (7) and grading function Equation (8) can be obtained as follows:

$$s(P_i(x, y)) = 0.45 \cdot mean(P_i(x, y)) + 0.35 \cdot \min(P_i(x, y)) + \\ 0.1 \cdot (\max(P_i(x, y)) + \min(P_i(x, y))) \tag{7}$$

where $s$ is the grading standard. Equation (8) is, specifically, as follows:

$$g(P(x, y)) = \begin{cases} 2, & when \ \text{sum}(I(x, y) > s(P_i(x, y))) > 5 \\ 0, & when \ \text{sum}(I(x, y) > s(P_i(x, y))) < 4 \\ 1, & ortherwise \end{cases} \tag{8}$$

where $g$ is the grade, $I(x, y)$ and is the pixel of the original image.

Such an observation can be used to repair the discontinuous foreground with the pixels and masks of Grade 2 or 1. The Grade 2 or 1 around a pixel is the same as the deep color pattern of the 24 masks described above. The convolution is used to map the values into the decision space, and a threshold is given to find the probably fractured continuous foreground.

The overall pulmonary airway circling action was divided into initial airway circling and detailed airway optimization [24]. The initial airway circling was implemented using the region growing to segment the images with complex edges [32]. It was used to search for the initial shape of the pulmonary airway. The region growing used the similar grayness of the same object in the image to collect pixels, and an initial seed was given. The adjacent pixels with differentiation within the threshold were brought into the region to grow until all pixels in the region were free of additional pixels. The similarity of the pixels in the region could be determined by the average degree of grayness, texture, and gradient. The process is expressed as the following Equations (9)–(11):

$$m = I(s) \cdot se \tag{9}$$

$$k(s \pm 1) = \begin{cases} 1, & when \ |m(s \pm 1) - I(s)| < d \\ 0, & when \ |m(s \pm 1) - I(s)| > d \end{cases} \tag{10}$$

$$g(s \pm 1) = g(s \pm 1) \cup (s \pm 1) \tag{11}$$

where $s$ is the seed, and it is a coordinate, $I$ is the image, $m$ is the decision space, $d$ is the decision condition, $k$ is the judgment result, and $g$ is the seed growing region.

If $k$ has a value of 1, the relative position of the point in $I$ is figured out. The position is set as $s$. Equations (9)–(11) are executed until $k$ is free of any value of 1.

The detailed airway optimization is implemented using the secondary 3D region growing method in the restricted area of the morphological gradient method [24]. However, the morphological gradient method not only finds out the gradient of the airway, but it will also probably find the gradient induced by blood vessels. Therefore, grayscale reconstruction methods of different cores were used to find the enclosed local dark pixels to optimize the result of the initial airway circling.

The operation of grayscale reconstruction can be regarded as continuous dilation or erosion of a gray level image [34]. It is similar to the hole-filling method, except the grayscale reconstruction has a labeled image that continuously dilates or erodes until its contour matches the original image. At this point, the grayscale reconstruction is completed. In order to connect the detection and combination, this study used the original image and the labeled image via a closed operation. To look for the valley, the labeled image was eroded repeatedly and compared with the original image until the labeled image did not change anymore. The closed local valley was looked for according to the difference between the labeled and the original images. The process is expressed as the following Equations (12)–(14):

$$J_{k=1} = (I \oplus B)\Theta B \tag{12}$$

$$J_{k+1} = \max(J_k \Theta B, I) \tag{13}$$

$$D = J_{\inf} - I \tag{14}$$

where $J$ is the labeled image, $J_k$ means $J$ went through $k$ iterations, $I$ is the original image, $B$ is the structuring element, $\oplus$ represents the dilation operator, $\Theta$ represents the erosion operator, and $D$ is a local difference for detecting the target.

## 2.4. Data Systematization

The implementation challenge is to avoid redundant computing data and, thus, the pulmonary airway mask should be reduced to an accessible format. Hence, a serial tree structure was used to access the information when naming and recognizing pulmonary bronchi.

Firstly, the skeleton of the airway mask is left over by morphological 3D skeletonization. In the skeletonized airway, the voxels around each voxel will determine the location of the voxel in the image, expressed as the following Equation (15):

$$
\begin{aligned}
N(x,y,z) &= \sum_{i=-1}^{1}\sum_{j=-1}^{1}\sum_{k=-1}^{1} S(x+i,y+j,z+k)-1, \\
G(x,y,z) &= connect\ node,\ when\ N(x,y,z) > 2, \\
G(x,y,z) &= line,\ when,\ N(x,y,z) = 2, \\
G(x,y,z) &= end\ node,\ when\ N(x,y,z) = 1,
\end{aligned}
\tag{15}
$$

where $N$ is the number of voxels around a voxel, and $S$ is the skeletonized 3D airway. Here, $x$, $y$, and $z$ are the spatial coordinates of a voxel, while G is the recording space. The connect node represents the place where an airway branches. The line represents an airway, and the end node represents the endpoint of an airway. When recording the nodes, all of the voxels should act as in the image.

The end node and connect node in the recording space G were used as the graph's points. The spatial distance of the plurality of lines between the end node and connect node was used as the edge point distance, and the graph could be completed. The cycles and the connect-nodes too close in the graph were removed. The highest end node in the recording space 'G' was selected as the root (upper part of the main trachea). The graph was then shaped into a tree. The data in the link series to be stored include the following:

(1) Father node—any node iterated father node in the tree that can point at the tree root to represent the airway source;
(2) Child node—any node iterated child node in the tree that can point at a tree bottom to represent the branch and link of the airway;
(3) Coordinate—marking the coordinates of the point in the space;
(4) Father distance—the distance to the father node, derived from the graph line cost of this node to the father node;
(5) Child distance—the distance to the child node, derived from the graph line cost of this node to the father node;
(6) Father vector—the direction of the father node, subtracting the coordinates of the father node from the coordinates of this node;
(7) Child vector—the direction of the child node, subtracting the coordinates of this node from the coordinates of a child node.

The anatomic branch names of all the airways can be identified based on the above dataset.

### 2.5. Bronchial Identification

The implementation challenge is, at present, that the bronchi are nominated according to their trends. To identify bronchi, the world coordinates about the patient are required instead of the slice space coordinates built of CT slices. The world coordinates are defined as follows:

(1) Z-axis—main trachea (upper) direction i.e., negative child node direction of a tree root;
(2) Y-axis—longitudinal direction, i.e., outer product of nodal coordinates of left airway minus nodal coordinates of the right airway to Z-axis;
(3) X-axis—horizontal direction, i.e., the outer product of the Y-axis and Z-axis.

Thereby, the directions concerning the patient can be obtained. After obtaining the world coordinates, all known airways could be compared, identified, and classified. This study used hierarchical classification [30] to identify anatomic Grade I, Grade II, and Grade III. Our identification conditions were derived from the patient's world coordinates and the currently included angle of branches. After identification, all of the child nodes subordinate to these defined nodes were defined as the same class. The airway mask was classified accordingly. The mask was redivided into masks nominated by different bronchi.

### 2.6. Result Display

This study used the marching cube algorithm [35] to extract the isosurface in order to reconstruct the 3D airway tree image. The algorithm assumed that the data were formed of discrete data points in 3D orthogonal space. Taking a CT image as an example, the serial 2D slices meet the condition of the algorithm. If the number of slices is 20 and the length and width of each slice are 512, the samples are taken 512, 512, and 20 times in x, y, and z directions, respectively. A function f(x, y, z) is used to obtain the continuous target slice image. The 3D image is reconstructed based on this algorithm.

The isosurface extraction algorithm confirms the voxels through the isosurface and uses unit blocks to build a triangular model. In terms of the discrete data points in 3D orthogonal space, eight adjacent data points can construct a unit block. The eight vertices of the unit block comprise two layers of CT image pixels. Each layer of pixels was connected to the defined f(x, y, z) function by the method, and the iso-value was given. The function f(x, y, z) = c created a surface. The region of intersection of the unit block and surface was obtained, and the nearest polygon was found.

The isosurface was deduced to a 3D dataset. The unit square was changed to a unit block and the vertices were changed from four pixels to eight voxels.

## 3. Experimental Results

The details and image processing results at various stages of the system proposed in this study are introduced in this section, including data preprocessing, lung circling, pulmonary airway circling, data systematization, bronchial identification, and the results.

### 3.1. Image Preprocessing

The CT images have interference and noise for various reasons. For this reason, the adaptive median filter was used for noise canceling, and the four neighbors low pass filter was used to enhance the pixels of the suspected airway wall, as shown in Figure 2. The noise was eliminated, and the overall sharpness was reduced.

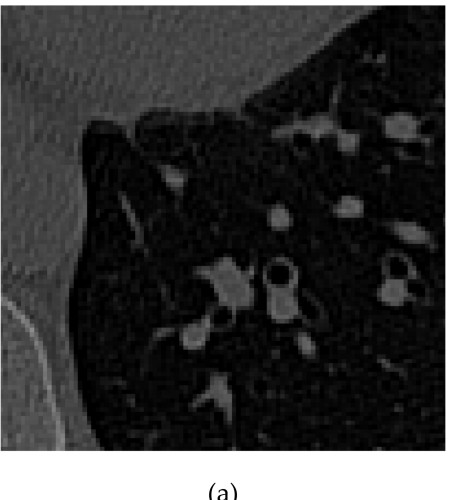
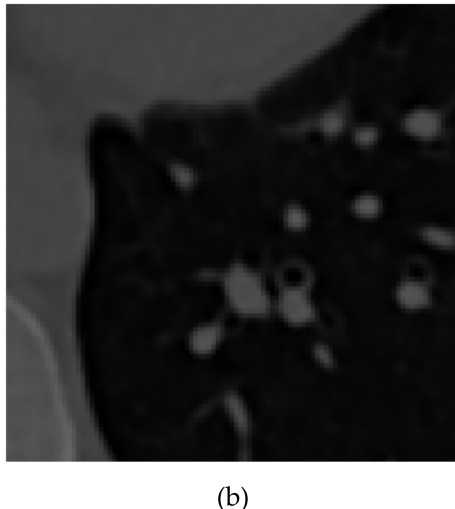

(a)　　　　　　　　　　　　　　　　　　　　(b)

**Figure 2.** Image contrast processing comparison diagram: (**a**) before processing (**b**) after processing.

### 3.2. Lung Circling

This study used k-means clustering to obtain the air (yellow in Figure 3) and quasi-air (light blue) pixels in the CT. The edge mask was removed by representation to complete the lung circling, and the internal voids were removed by morphology, as shown in Figure 4.

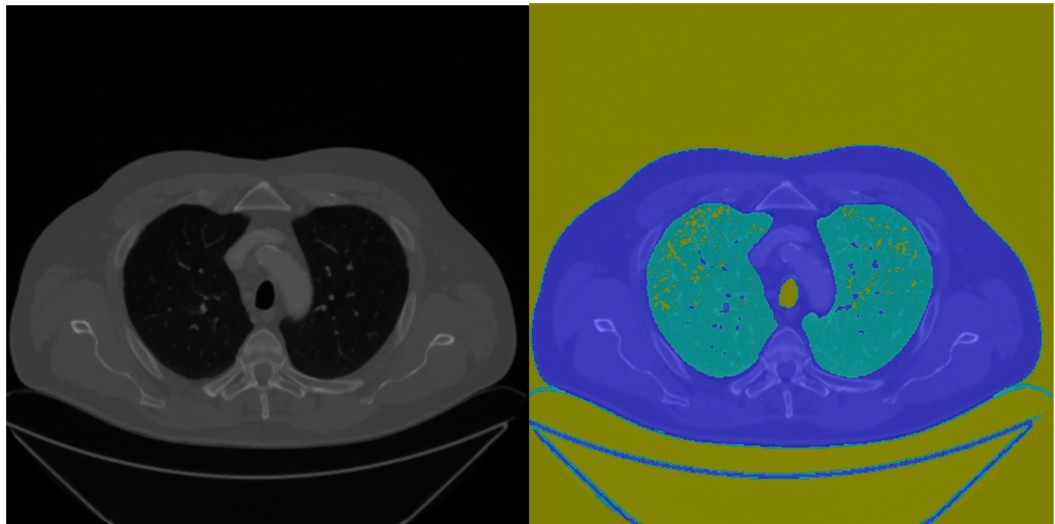

**Figure 3.** K-means initially segmented image.

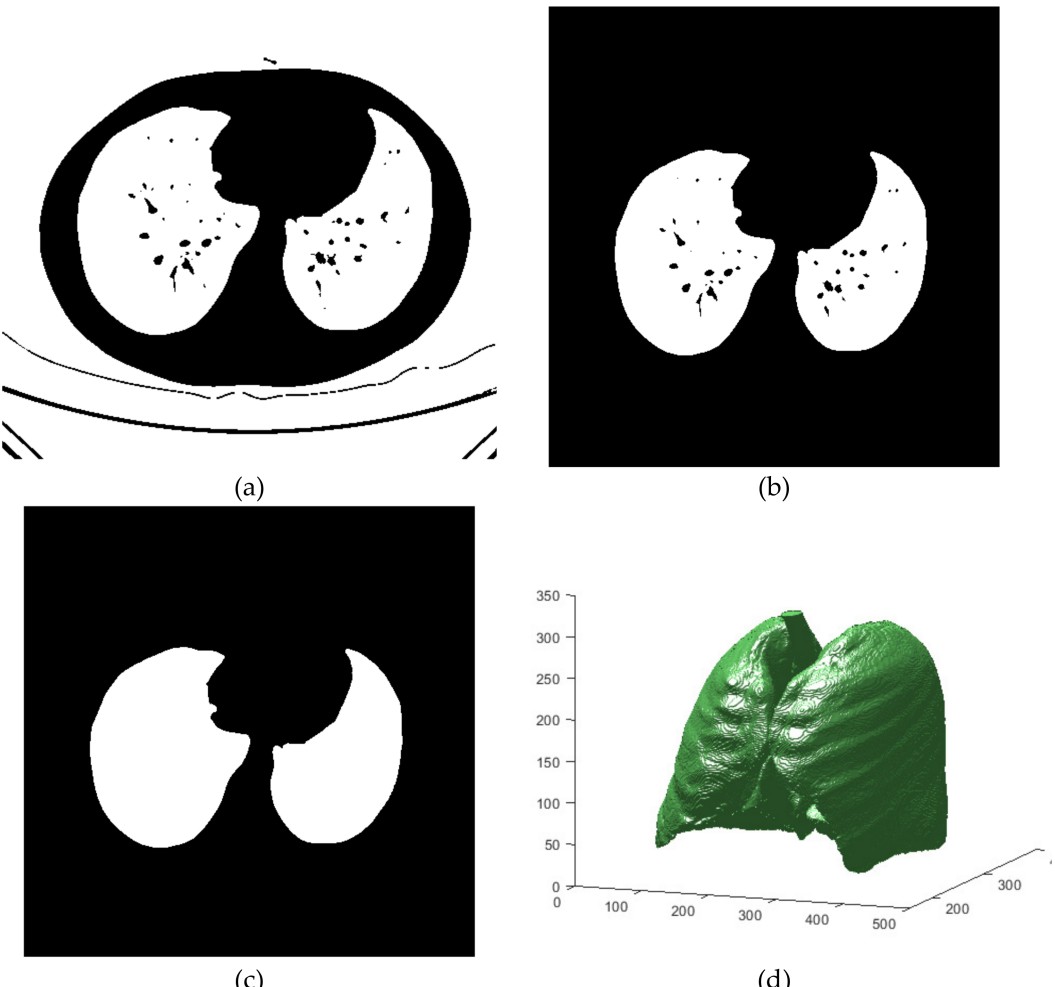

**Figure 4.** Lung mask extraction. (**a**) Binary K-means result; (**b**) remove edges and small objects; (**c**) 3D closed operation result; (**d**) 3D reviewed lung mask extraction result.

### 3.3. Pulmonary Airway Circling

Due to the restrictions of CT and the effects of diseases and noise, the pulmonary airway wall is sometimes fractured or injured. It should be repaired before the pulmonary

airway is circled. Each pixel of the lung mask is given a score to assess how much the pixel looks like an airway (blood vessel). The pixel of a probably ruptured airway wall is judged according to the plane's position distribution of pixel scores, as shown in Figure 5.

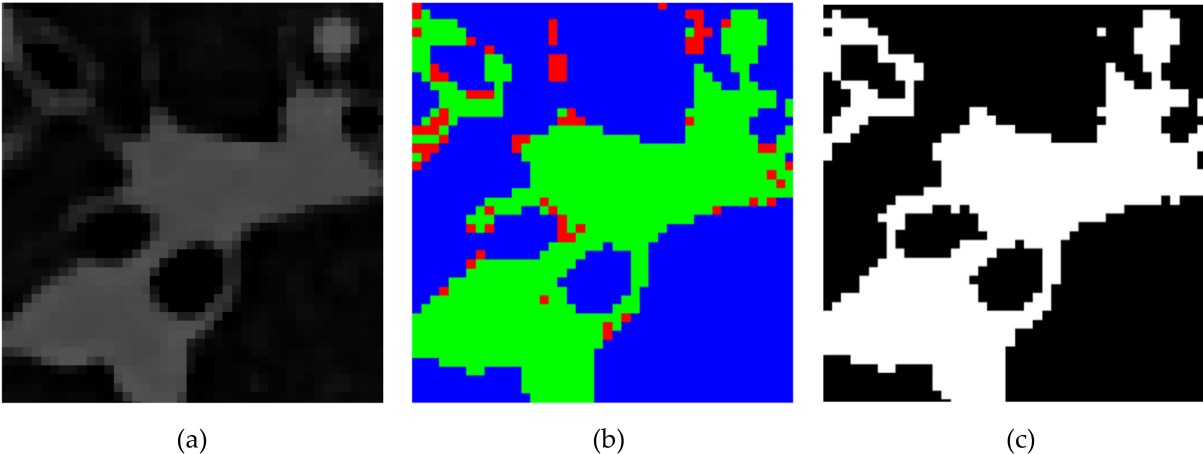

(a)                                          (b)                                          (c)

**Figure 5.** Pulmonary airway wall repair. Here, (**a**) is the original image, (**b**) is the score given according to three-stage segmentation; blue (air) is 0 point, red (uncertain) is 1 point, and green (airway wall and blood vessel structures) is 2 points, and (**c**) is the result of repair after the mask weight score is applied.

The uncertain pixels are dualized to air or not air to generate a mask. The pixel values in the original image are changed to a K-means tissue part and other object cluster centers of high radiation shielding according to the mask to complete filling.

After filling, the closed dark pixels are detected by the grayscale reconstruction method, as shown in Figure 6. Grayscale reconstruction masks of different detection radii are connected using a six-adjacency relation to complete pulmonary airway detection. Figure 7 shows the 3D visualization result of the circled airways.

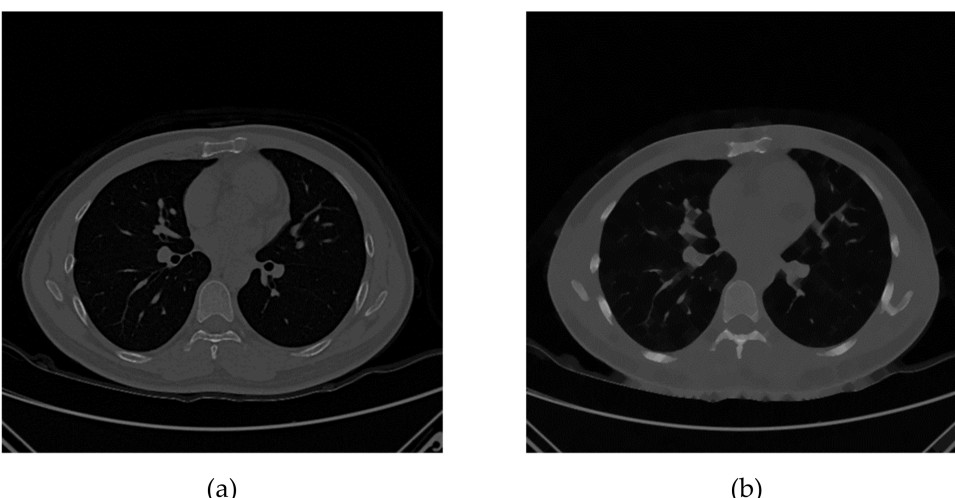

(a)                                          (b)

**Figure 6.** *Cont.*

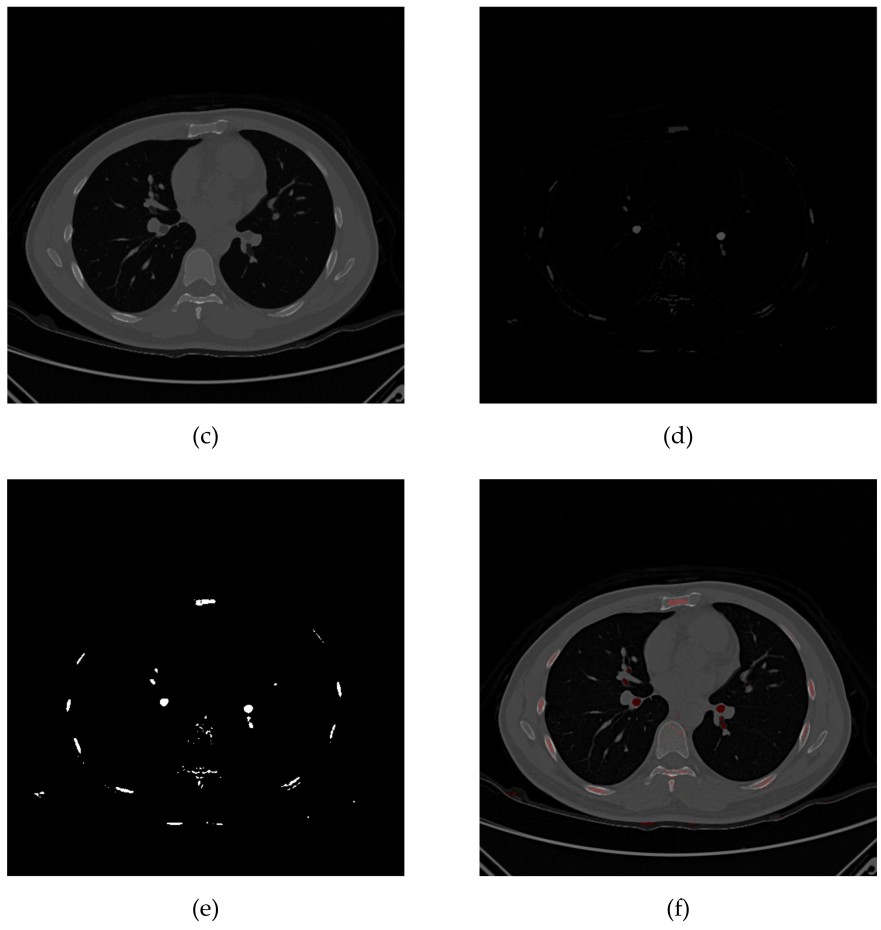

Figure 6. Grayscale reconstruction flow chart. (**a**) Original image; (**b**) labeled image; (**c**) grayscale reconstruction image; (**d**) local difference image; (**e**) local valley target; (**f**) original image marked with local valley target.

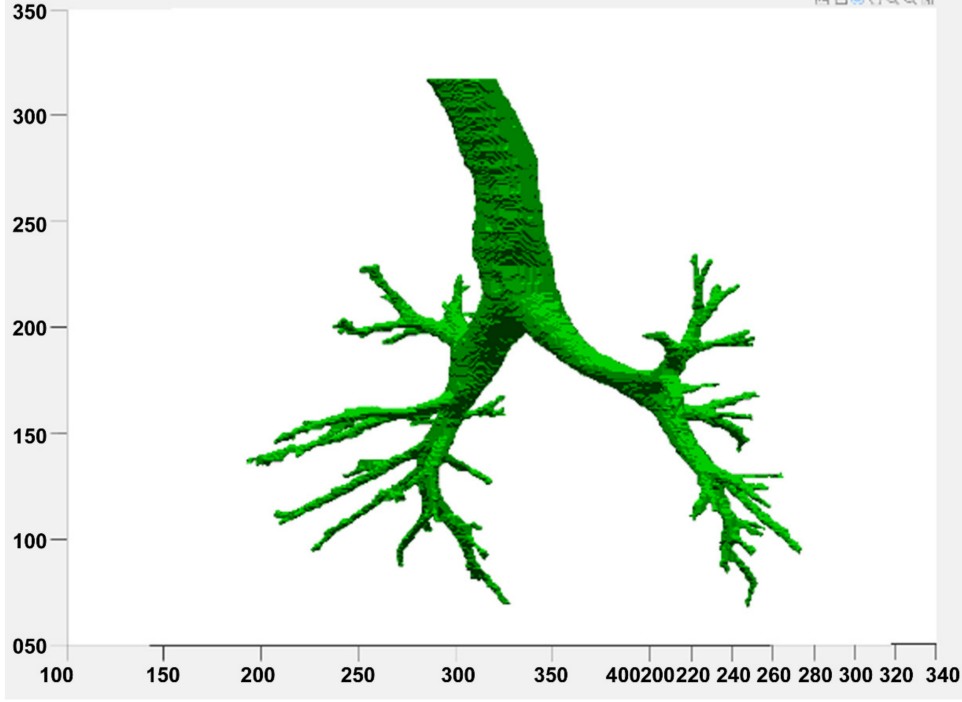

**Figure 7.** 3D reviewed pulmonary airway extraction result.

### 3.4. Data Systematization

The trend information is enough for identifying the bronchus name. To reduce nonrequired information and accelerate identification, the airways should be shaped into a tree using the method in Section 2.4, as shown in Figures 8 and 9.

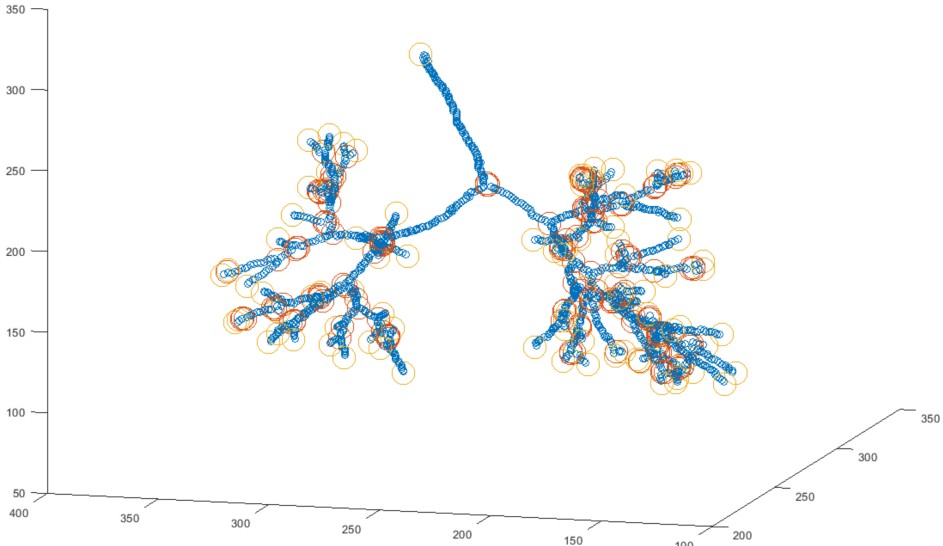

**Figure 8.** Airway skeleton drawing. Here, orange is a node and blue is a line.

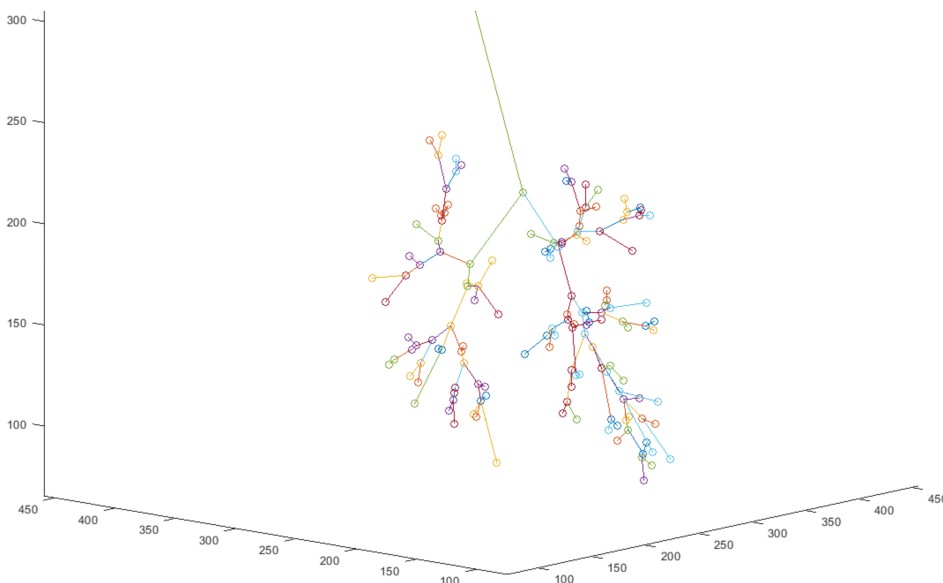

**Figure 9.** Visual tree structure.

### 3.5. Bronchial Circling

Based on the relationship between the superior and subordinate of the tree structure and the patient's world coordinate angle comparison, the tree structure can be labeled, as shown in Figure 10. The pulmonary airway mask at the shortest Euclidean distance is attributed to the labeled tree structure. Labeled 3D pulmonary airways are shown in Figure 11.

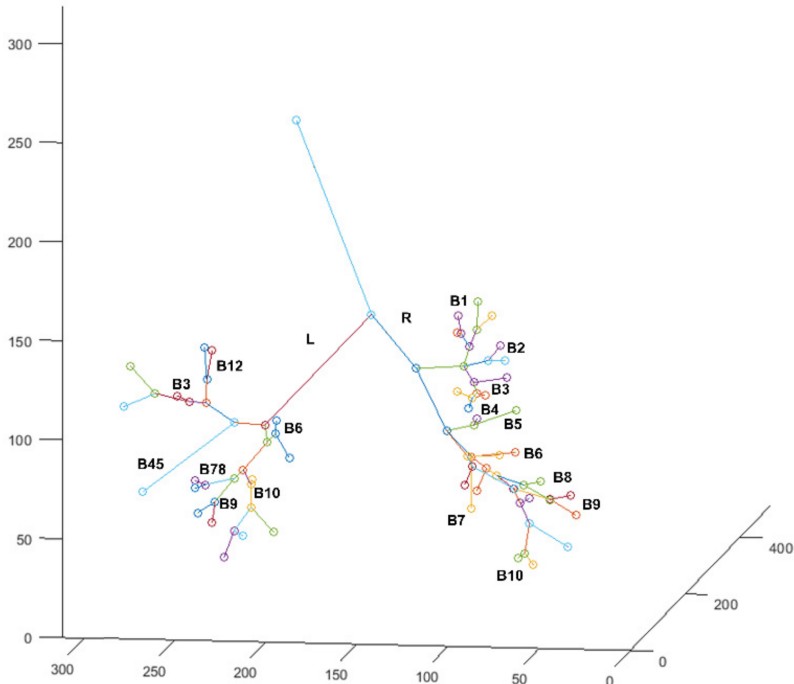

**Figure 10.** Labeled tree structure.

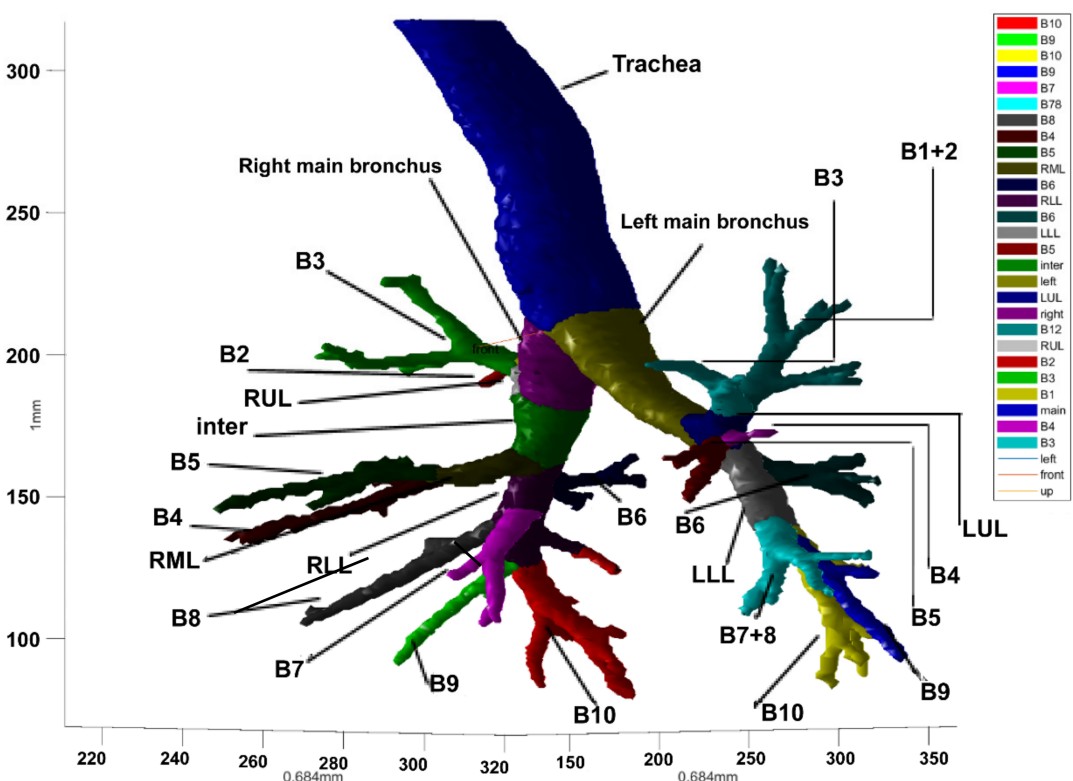

**Figure 11.** Labeled pulmonary airways (45 degrees).

### 3.6. Result Display

This system enables the user to look for arbitrary points in the lungs of interest in the 2D graphics files. The system compares all of the end nodes and extracts the point at the shortest Euclidean distance as the attribution of the coordinates. Then, it returns to the tree root by inverse iteration to list the most probable paths of the coordinates to the main

trachea through the airways, which are mapped into the 3D graphics file. As shown in Figure 12, the average time for identifying each point is 2.11 s.

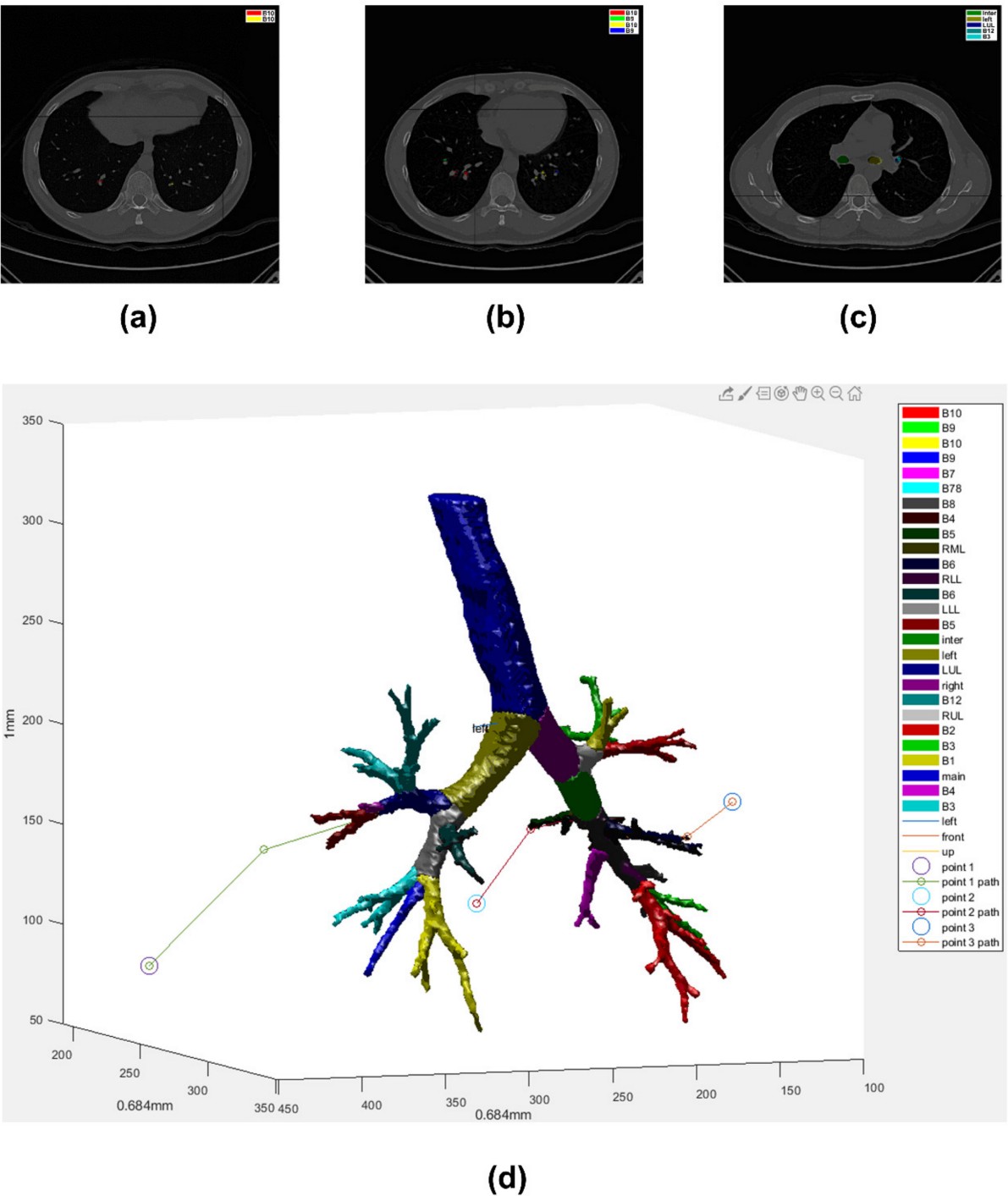

**Figure 12.** Attribution test. (**a**) Test point 1; (**b**) test point 2; (**c**) test point 3; (**d**) 3D graphics file mapping result.

*3.7. Airway Circling Method*

To automate the workflow without predetermined data to the maximum extent, this study used the architecture of the secondary region growing method. It changed the morphological gradient restriction of the secondary region growing method to a multi-radius grayscale reconstruction. The detection radius was adjusted dynamically to refine the pulmonary airway block derived from the primary region growing method. Additionally,

as the local closed valley was detected using the grayscale reconstruction method, the closure of the region was important. The discontinuous foreground (highland) repair of the three-stage segmentation was imported.

The secondary region growing method, grayscale reconstruction method, and three-stage segmentation method were used. This study tried to reconstruct a model for pulmonary airway detection. It was compared with the method proposed in this study. Table 1 shows the topological algebra and quantity detected by our proposed method. Figure 13 visually demonstrates the topological algebra and quantity detected by different pulmonary airway detection algorithms.

**Table 1.** Topological algebra and quantity detected by our method.

| File | Order of Bronchial Tree Division by Our Method | | | | | | | | | | | |
|---|---|---|---|---|---|---|---|---|---|---|---|---|
|  | 1st | 2nd | 3rd | 4th | 5th | 6th | 7th | 8th | 9th | 10th | 11th | All |
| file1 | 1 | 2 | 4 | 9 | 18 | 14 | 12 | 6 | 6 | 2 | 0 | 74 |
| file2 | 1 | 2 | 4 | 8 | 17 | 21 | 6 | 7 | 5 | 2 | 2 | 75 |
| file3 | 1 | 2 | 4 | 9 | 21 | 25 | 27 | 19 | 15 | 4 | 2 | 129 |
| file4 | 1 | 2 | 4 | 8 | 18 | 11 | 10 | 4 | 5 | 2 | 2 | 67 |
| file5 | 1 | 2 | 4 | 8 | 16 | 28 | 13 | 7 | 6 | 2 | 0 | 87 |
| file6 | 1 | 2 | 4 | 9 | 20 | 24 | 26 | 14 | 6 | 1 | 0 | 107 |
| file7 | 1 | 2 | 4 | 9 | 17 | 16 | 4 | 4 | 2 | 1 | 0 | 60 |
| file8 | 1 | 2 | 4 | 9 | 18 | 22 | 6 | 2 | 1 | 0 | 0 | 65 |
| file9 | 1 | 2 | 4 | 8 | 18 | 26 | 16 | 5 | 4 | 2 | 0 | 86 |
| file10 | 1 | 2 | 4 | 9 | 18 | 19 | 12 | 6 | 6 | 4 | 0 | 81 |
| Average | 1 | 2 | 4 | 8.6 | 18.1 | 20.6 | 13.2 | 7.4 | 5.6 | 2 | 0.6 | 83.1 |

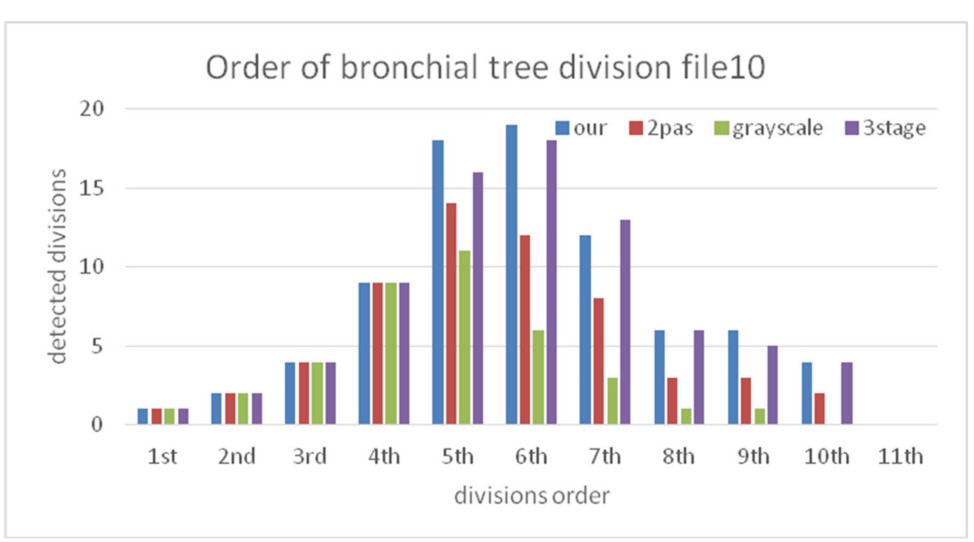

**Figure 13.** Comparison diagram of detected bronchi.

As shown in Figure 13, there is no difference among the four methods in recognition of the first generation in topology of the main trachea, the second generation of the left and right main bronchi, or the third generation of the left upper lobe bronchus, left lower lobe bronchus, right upper lobe bronchus, and right middle bronchus. In the fourth-generation trachea search, as it is in the tertiary branch of anatomy, the performance shows differences among files. This is because the tertiary branch of anatomy may have individual differences. In addition, different algorithms have detection differences in processing airways near a tracheal wall for the differences induced by the algorithm. Therefore, the structured system has differences in identifying topological algebra. For example, algorithm A detects one more pixel toward the airway wall than algorithm B in airway detection. This leads to the differences, as shown in Figure 14.

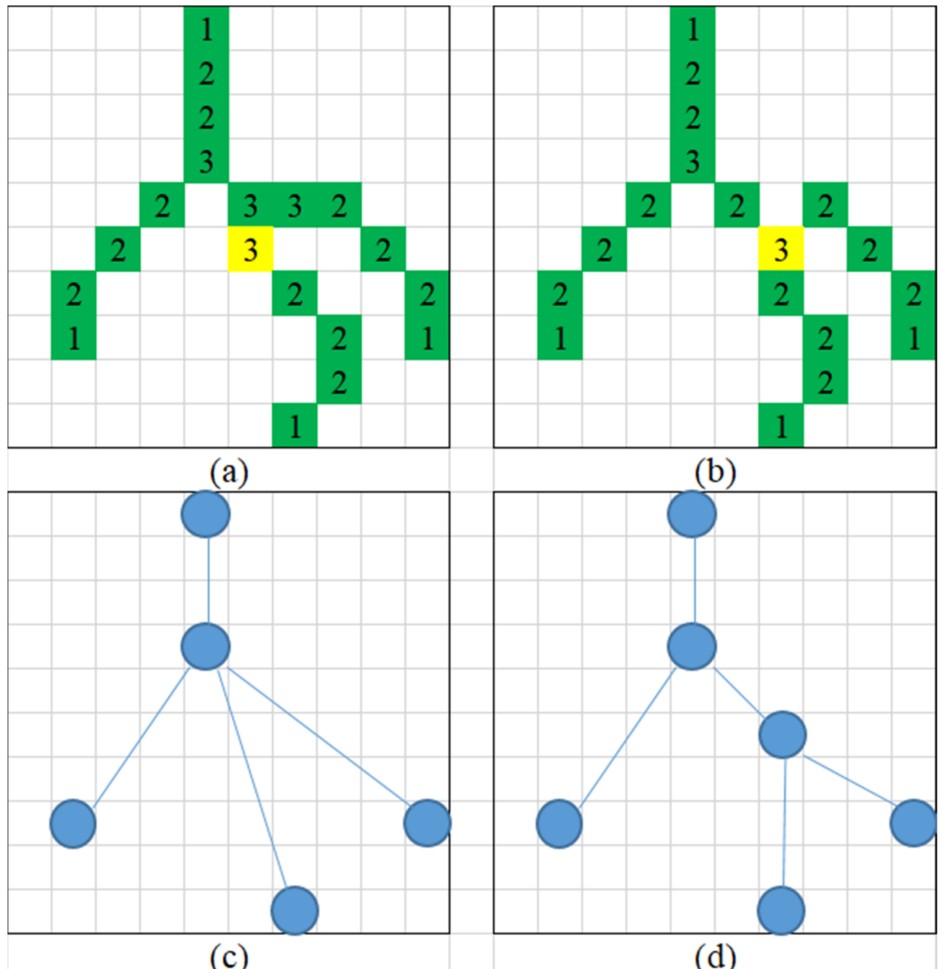

**Figure 14.** An example of the algebraic difference in airways. Here, (**a**) is the skeletonization example of the assumed algorithm A, (**b**) is the skeletonization example of the assumed algorithm B, (**c**) is the tree diagram converted from (**a**), and (**d**) is the tree diagram converted from (**b**).

When the branches enter the fifth-generation bronchi in topology, the main tracheas of the left upper and right upper lobe enter the quaternary branches. The left lower, and right middle begins to enter the tertiary branches. It is difficult to recognize the influencing airway. However, when entering the quaternary branches, the differences among algorithms begin to occur. In most cases, the number of airways found by the pure grayscale reconstruction method has peaked. It begins to decrease after the sixth generation. When the branches have entered the sixth-generation bronchi in topology, nearly all of the algorithms have passed the peak, because the subordinate branches of the left upper and right upper lobe bronchi have been unable to find finer bronchi. Finally, the algorithm proposed in this study can find the tenth-generation bronchi in most cases and find the eleventh-generation bronchi in the optimal case.

### 3.8. Bronchial Identification and Lung Segmentation

After the pulmonary airway detection was completed, the anatomical name of the pulmonary airway could be identified. This study performed data systematization and obtained the world coordinates subject to the patient. The average time from pulmonary airway masking to completion of bronchial identification was only 1.646 sec. Furthermore, a professional radiologist judged the recognition results of this study. Table 2 shows the judgment result. Here, 2 points means the bronchus is found and correct, 1 point means the bronchus is found, but the result is a little defective, and 0 points means that the recognition is incorrect or failed.

**Table 2.** Bronchial identification evaluation.

| File | | B1 | B2 | B3 | B4 | B5 | B6 | B7 | B8 | B9 | B10 |
|---|---|---|---|---|---|---|---|---|---|---|---|
| file1 | right | 2 | 2 | 2 | 2 | 2 | 2 | 2 | 2 | 2 | 2 |
| | left | | 2 | 2 | 2 | 2 | 2 | | 2 | 2 | 2 |
| file2 | right | 2 | 2 | 2 | 2 | 2 | 2 | 2 | 2 | 2 | 2 |
| | left | | 2 | 2 | 2 | 2 | 2 | | 2 | 1 | 1 |
| file3 | right | 2 | 2 | 2 | 2 | 2 | 2 | 2 | 2 | 2 | 2 |
| | left | | 2 | 2 | 2 | 2 | 2 | | 2 | 2 | 2 |
| file4 | right | 2 | 2 | 2 | 2 | 2 | 2 | 2 | 2 | 2 | 2 |
| | left | | 2 | 2 | 2 | 2 | 2 | | 1 | 2 | 2 |
| file5 | right | 2 | 2 | 2 | 2 | 2 | 2 | 2 | 2 | 2 | 2 |
| | left | | 2 | 2 | 2 | 2 | 2 | | 2 | 2 | 2 |
| file6 | right | 2 | 2 | 2 | 2 | 2 | 2 | 2 | 2 | 2 | 2 |
| | left | | 2 | 2 | 2 | 2 | 2 | | 2 | 2 | 2 |
| file7 | right | 2 | 2 | 2 | 2 | 2 | 2 | 2 | 2 | 2 | 2 |
| | left | | 2 | 2 | 2 | 2 | 2 | | 2 | 2 | 2 |
| file8 | right | 2 | 2 | 2 | 2 | 1 | 2 | 2 | 2 | 2 | 2 |
| | left | | 2 | 2 | 2 | 2 | 2 | | 2 | 2 | 2 |
| file9 | right | 2 | 2 | 2 | 2 | 2 | 2 | 2 | 2 | 2 | 0 |
| | left | | 2 | 2 | 2 | 2 | 2 | | 2 | 2 | 2 |
| file10 | right | 2 | 2 | 2 | 2 | 2 | 2 | 2 | 2 | 2 | 2 |
| | left | | 2 | 2 | 2 | 2 | 2 | | 2 | 2 | 2 |

As shown in Table 2, our bronchial identification has very high accuracy. The part identified as 1 point is from partial mapping errors of the mask, as confirmed by a radiologist. Most of the labels are correct, and the scoring rate is 0.983.

In intrapulmonary localization, an arbitrary point in the lungs is connected to the known nearest bronchus. The success rate and accuracy of the method are based on the algebra of bronchial identification. The more detailed bronchi means/results from a higher localization accuracy. Theoretically, such a Euclidean spatialization method performs navigation by assuming unidentifiable capillary bronchus paths. If the navigation is executed for all the pixels in the lung mask, Figure 15 can be obtained.

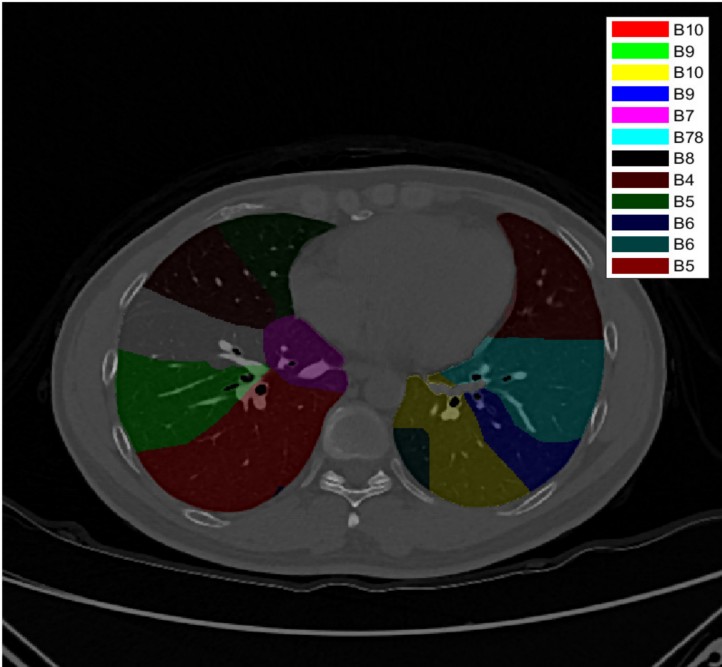

**Figure 15.** Navigation result.

As shown in Figure 15, the attribution classification result is approximately reasonable. However, if the classification has distinguishable pulmonary fissures, the boundary between different labels does not account for the fissures, which are suspected of having been caused by an insufficient quantity of detected fine bronchi. The virtual airway formation theory proposed by Nousias et al. [36] may help to solve the problem, but they did not propose the relationship of virtual airway formation to pulmonary fissures.

### 3.9. Airway Parameter Measurement and 3D Printing Verification

After completing trachea reconstruction, the linear interpolation x, y, and z axes of the mask are orthogonally masked by the space in the unit of 1 mm, as shown in Figure 16. It is converted into a stl file. The recognized pulmonary airways are reproduced by SLA technology for doctors' reference, as shown in Figure 17. Compared to the original 2D CT or the 3D graphics files in the computer, the 3D printed pulmonary airways are more convenient for doctors. This allows the doctors to check the regions of the airways, and such models can authentically display the relative positions of intrapulmonary localization.

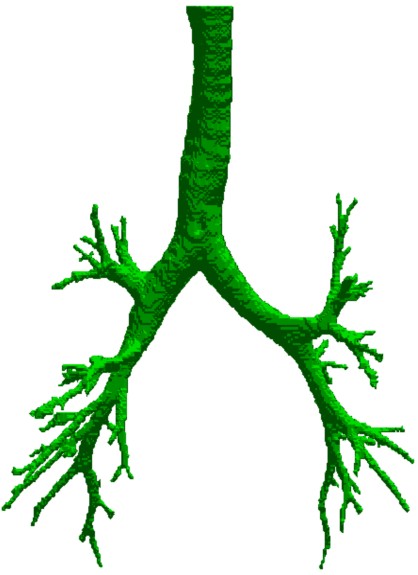

**Figure 16.** Corrected pulmonary airways.

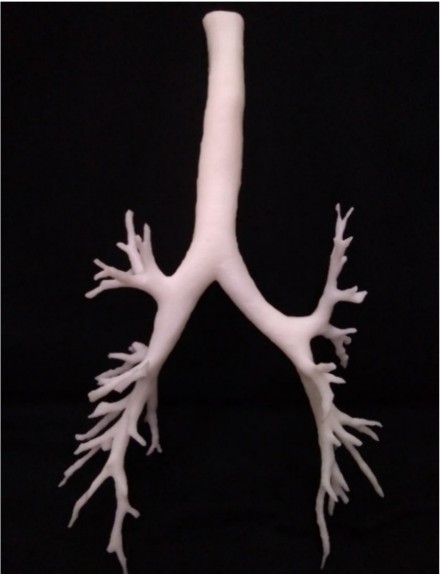

**Figure 17.** 3D printed pulmonary airways.

The pulmonary airway parameters, such as length, diameter, and volume, are vital for preoperative planning. This study measured the detected main trachea. The measuring method is outlined in Figure 18. The specific measured parameters include the following:

(1) Length of the main trachea—length from the larynx to the bifurcation of the main trachea;

(2) Length of the left main bronchus—length from the bifurcation of the main trachea to the leftmost lower bronchus;

(3) Length of the right main bronchus—length from the bifurcation of the main trachea to the rightmost lower bronchus;

(4) The cross-sectional area of the main bronchus junction—cross-sectional area of the main bronchus, taking its trend as a normal vector at the airway bifurcation;

(5) The cross-sectional area of the left main bronchus junction—cross-sectional area of the left main bronchus, taking its trend as a normal vector at the airway bifurcation;

(6) The cross-sectional area of the right main bronchus junction—cross-sectional area of the right main bronchus, taking its trend as a normal vector at the airway bifurcation;

(7) The main bronchus's cross-sectional area—cross-sectional area of the main bronchus, taking its trend as a normal vector (multi-section average);

(8) The cross-sectional area of the left main bronchus—cross-sectional area of the left main bronchus, taking its trend as a normal vector (multi-section average);

(9) The cross-sectional area of the right main bronchus—cross-sectional area of the right main bronchus, taking its trend as a normal vector (multi-section average);

(10) Diameter of the main bronchus—long diameter and short diameter of the cross-sectional area of the main bronchus, taking its trend as a normal vector (multi-section average);

(11) Diameter of the left main bronchus—long diameter and short diameter of the cross-sectional area of the left main bronchus, taking its trend as a normal vector (multi-section average);

(12) Diameter of the right main bronchus—long diameter and short diameter of the cross-sectional area of the right main bronchus, taking its trend as a normal vector (multi-section average);

(13) Perimeter of the main bronchus—perimeter of the cross-sectional area of main bronchus, taking its trend as a normal vector (multi-section average);

(14) Perimeter of the left main bronchus—perimeter of the cross-sectional area of left main bronchus, taking its trend as a normal vector (multi-section average);

(15) Perimeter of the right main bronchus—perimeter of the cross-sectional area of right main bronchus, taking its trend as a normal vector (multi-section average);

(16) Angle of the left main bronchus—deviation angle of a left main bronchus from the main bronchus;

(17) Angle of the right main bronchus—deviation angle of a right main bronchus from the main bronchus;

(18) Volume of the main bronchus—spatial volume occupied by the main bronchus;

(19) Volume of the left main bronchus—spatial volume occupied by the left main bronchus;

(20) Volume of the right main bronchus—spatial volume occupied by the right main bronchus.

These parameters measure the main trachea and the left and right main bronchi. It is observed that the trends of cross-section area and angle conform to general airway law, except for the long diameter. This is because the left main bronchus is relatively elliptic. Taking tracheostomy as an example, the provided diameter of the main trachea is helpful in the selection of tracheostomy tube diameter. Additionally, the right and left main bronchi diameters can assist in a tracheal endoscopy. In addition, as the actual pulmonary airway parameter measurement is unavailable, 3D printing is performed for a group of data to verify the accuracy. This is shown in Table 3.

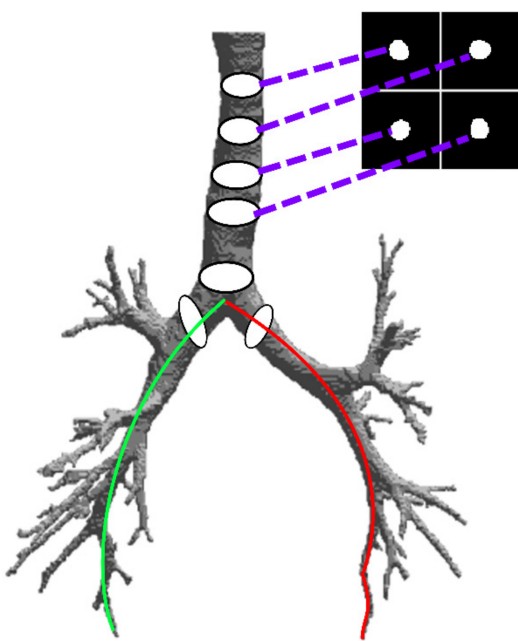

**Figure 18.** Measurement method (the redline is the length of the left main trachea, while the green line is the length of the right main trachea).

**Table 3.** 3D printing verification.

|  | **System Measurement** | **3D Printing** | **Accuracy** |
|---|---|---|---|
| Length of main trachea (mm) | 98.16 | 100.4 | 0.98 |
| Length of left main bronchus (mm) | 167.41 | 168.1 | 1.00 |
| Length of right main bronchus (mm) | 161.46 | 161.9 | 1.00 |
| Long/short diameter of main bronchus (mm) | 16.21/14.2 | 16.20/14.00 | 0.99/0.99 |
| Long/short diameter of left main bronchus (mm) | 10.41/7.87 | 10.42/7.82 | 1.00/0.99 |
| Long/short diameter of right main bronchus (mm) | 8.65/7.89 | 8.63/7.77 | 0.99/0.98 |
| Perimeter of main trachea (mm) | 45.69 | 44.9 | 0.98 |
| Perimeter of left main trachea (mm) | 26.63 | 26.3 | 0.99 |
| Perimeter of right main trachea (mm) | 24.58 | 24.4 | 0.99 |
| Angle of left main bronchus (degrees) | 44.2 | 44.5 | 0.99 |
| Angle of the right main bronchus (degrees) | 32.11 | 32 | 1.00 |

As shown in Table 3, 3D printing is excellent for verifying accuracy. The accuracy is higher than 98%.

## 4. Discussion

Pulmonary airway recognition is used extensively in preoperative planning evaluation and preliminary work for endoscope navigation. However, most studies did not mention bronchial identification. The pulmonary airway recognition and bronchial identification have been studied, but no vertically integrated system has been published.

In recent years, machine learning has been extensively used in pulmonary airway segmentation operations, as shown in Table 4. Bian et al. [35], Cheng et al. [36], Lee et al. [37], and Qin et al. [38] showed that machine learning had superior performance in pulmonary airway recognition. However, the performance of machine learning sometimes depends on the preparation of training data, and mass predetermined data means a large consumption of manpower. Lee et al. [37] indicated that the labeling of a group of data should be handled by professionals for 2~4 h. Better results cannot be obtained until 55 groups of data are labeled. In contrast, the research of Meng et al. [39], Nardelli et al. [40], and Gil et al. [41] did not require predetermined data and can be used directly, but the recognition accuracy is not as good as machine learning. Additionally, none of these studies

addressed bronchial identification. In the absence of predetermined information, this study has the best tree length performance, the shortest identification time, and a very high accuracy rate in bronchial identification.

**Table 4.** Application, systems approach, and effect of bronchial identification system.

| Method | Systems Approach | Path Length (mm) | Training Data | Airway Recognition Execution Time | Bronchial Identification |
|---|---|---|---|---|---|
| This study | Region growing, three-stage segmentation, grayscale reconstruction, secondary region growing, tree structure level recognition | 913 | None | 10~20 min | 98.3% |
| Bian et al. [37] | Hessian matrix feature, Random forest learning | max: 2895 min: 397 | 80 groups | Training: 2 h Prediction: 15 min | N/A |
| Cheng et al. [38] | Tiny atrous convolutional network (TACNet) | 1869 | 80 groups | N/A | 85.6% |
| Lee et al. [39] | Hybrid enhanced filtering (tubular detection + black hat transformation), fuzzy connection, SVM | 1217 | 55 groups | 10–30 min | N/A |
| Qin et al. [40] | Attention distillation aid U-net | 907 | 90 groups | N/A | N/A |
| Meng et al. [41] | Tubular detector | 559 | None | 4~5 h | N/A |
| Nardelli et al. [42] | Semiautomatic algorithm, manual seed regrowth | 751 | Semi-automatic | Semi-automatic | N/A |
| Gil et al. [43] | Pooling layer multiscale single diameter tubular detector, reverse skeletonization growth | 745 | None | 21.36 min | N/A |

In Table 4, the airway recognition performance was judged according to the tree length. The path length was calculated by adding the lengths of all of the found airway center lines, of which the identified airway depth could be represented visually. As shown in Table 4, besides the method of this study, the system of Cheng [38] provided the bronchial identification function while identifying airways, but the accuracy was not high. This study had the best tree length in the case without predetermined information, the shortest recognition time, and very high accuracy in bronchial identification.

In the surgical treatment of lung cancer, anatomic segmentectomy has the same effect as traditional lobectomy in eradicating cancer, but is superior in preserving lung function after surgery. When performing anatomic segmentectomy, the surgeon must have a clear understanding of the location of the target tumor and its surrounding anatomy. In order to reduce intra-operative complication and achieve precise resection, preoperative planning or intra-operative navigation with a 3D reconstruction have emerged and been widely applied in thoracic surgery. With the advancement of image processing technology, 2D images can be converted into 3D images, so as to effectively evaluate the bronchial branching pattern, discover the anatomical variation, determine the location of lesions, and clarify the division of the segments. This study also showed similar results for the morphometrics of human trachea and principal bronchi as per traditionally calculated morphometric data of human trachea [44]. Therefore, this study will contribute to lesion localization, preoperative simulation, the formulation of individualized surgical plans, and intraoperative navigation. We pushed the boundaries even further in the application of 3D CT reconstruction. Our modification will help play a positive role in anatomic pulmonary segmentectomy. At same time, the importance of the current study was that our results provide a quantitative measurement of the normal human trachea structure. The results of these findings will be useful for medical teaching and research. They will provide surgeons and anesthetists with tracheal-related information.

## 5. Conclusions

Since the airway's tree structure is complex, airway segmentation and bronchial recognition using CT are the keys to analyzing lung lesions. The micro-bronchial structure is subject to a partial volume effect. Due to the limited image intensity contrast among air, blood, and tissues, the difficulty level of segmentation is increased. Therefore, establishing a complete airway tree and bronchial recognition is very challenging, and this is the objective of this study. The conclusions of this study are as follows:

(1) This proposed an image processing method for automatic pulmonary airway detection and the bronchial recognition of chest cavity CT could be used to obtain a 3D model applicable to virtual bronchoscopy. The lesion path and intrapulmonary localization could be explored and planned through the proposed method;

(2) The pulmonary airway was displayed using 3D printing. Proper critical physiological parameters of bronchus were the criteria for diagnosis or performing airway disease operations. Twenty pulmonary airway parameters, including airway length, diameter, volume, carina junction angle, cross-sectional area, and cross-sectional area of carina junction were measured. The accuracy is higher than 98%.

(3) This developed pulmonary bronchus identification system for thoracic CT provides more lung information to doctors and shortens the data checking time. This system can segment and reconstruct airways automatically.

(4) The proposed model can search to the eleventh-generation bronchial segments without training for an airway search. The bronchial structure is recorded by a linked list, and the bronchus names are identified according to the world coordinates. The bronchial identification accuracy is 98.3%.

(5) The arbitrary points of the lungs are guided based on bronchi to calculate the most probable trachea of the arbitrary points in the lungs, assisting doctors in understanding regions of interest to increase diagnosis efficiency.

**Author Contributions:** C.F.J.K.: Methodology, Project administration, Writing—review & editing. Z.-X.Y.: Conceptualization, Methodology, Data curation, Investigation, W.-S.L.: Data curation, Formal analysis, S.-C.L.: Writing, Methodology, Supervision, Writing—review & editing. All authors have read and agreed to the published version of the manuscript.

**Funding:** This research received no external funding.

**Institutional Review Board Statement:** The study was conducted in accordance with the Declaration of Helsinki, and approved by the Institutional Review Board (or Ethics Committee) of Tri-Service General Hospital (protocol code C202105070 and Approval date: 25 May 2021).

**Informed Consent Statement:** Consent to publish has been obtained from all participants.

**Data Availability Statement:** Not Applicable.

**Acknowledgments:** The research was supported by the Tri-Service General Hospital, National Defense Medical Center, National Defense Medical Center-National Taiwan university of Science and Taichung Armed Forces General Hospital, and National Taiwan University of Science and Technology Joint Research Program (TSGH-A-111004, TCAFGH-E111044, TSGH-NTUST-111-03). The funders had no role in study design, data collection and analysis, decision to publish, or preparation of the manuscript.

**Conflicts of Interest:** The authors declare no conflict of interest.

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
