# Peer review of "Application of Image Processing and 3D Printing Technique to Development of Computer Tomography System for Automatic Segmentation and Quantitative Analysis of Pulmonary Bronchus"

_mathematics, doi:10.3390/math10183354_

Round 1
Reviewer 1 Report
In this paper, authors proposed an image processing method for automatic pulmonary airway detection and bronchial recognition to obtain a 3D model applicable to virtual bronchoscopy. The lesion path and intrapulmonary localization could be explored and planned through this method. Overall, this is a good topic work, and I agree with many of the conclusions that the authors draw but paper is not upto the standards of journal in terms of its presentation. The following review comments are recommended, and the authors are invited to explain and modify.
Comment: The title does not make much sense.
Comment: The abstract section is inconsistent and does not reflect the main contributions of the manuscript.
Comment: However, the manuscript is overly lengthy and some parts could be shortened or removed.
Comment: What is importance and application of automatic pulmonary bronchus segmentation, identification and quantitative analysis of Chest Computed Tomography?
Comment: Novelty is confusing. A highlight is required. The main contributions of the manuscript are not clear. The main contributions of the article must be very clear and would be better if summarize them into 3-4 points at the end of the introduction.
Comment: There are lots of typos. English needs to revise again and also the figures are not clear in some cases.
Comment: Nothing is mentioned about the implementation challenges.
Comment: The following clinical decision support systems and medical imaging must be included to improve the quality of the paper.
· 10.1155/2022/2665283
· 10.3390/math10050796
Comment: However, I have some concerns related to the difference with respect to the state-of-the-art and performed experimental results and comparison. It is not clear the improvements with other cited related works. Authors must do a comparative study with the state of the art.
Comment: Could you please check your references carefully (in particular, proceedings: location of the conference, date of the conference, publisher's name and location...)? All references must be complete before the acceptance of a manuscript.
Author Response
Please see the attachment (Response 1)

Reviewer 2 Report
The title of the article fully reflects the intent and content of the article.
The section "Abstract" contains the necessary information for the reader: relevance, an image processing method for automatic detection of the pulmonary airways and bronchial recognition to obtain a three-dimensional model applicable to virtual bronchoscopy, results are presented. The results are too extensive and overload the section. The conclusion is clear and corresponds to the results of the study. The proposed system retains the advantages of automation and high accuracy, contributes to clinical diagnosis and is the basis for improving treatment.
All keywords are necessary and reflect the research topic presented by the authors.
In the "Introduction" section, the authors presented the relevance of the study for lung diseases, lung cancer was isolated. Various methods of studying the lungs of patients presented in the literature and used in practice are characterized. The basics of these methods, advantages and disadvantages are described. This data is significant. The authors pointed out the need to develop a more advanced approach to lung research. At first glance, the presented picture of the methods is redundant, but further reading of the article shows that such a decision by the authors is advisable. The aim of the study is clearly presented, which is to develop an objective and accurate system of image processing methods for analyzing the structure of the lungs and providing information about the position between the lung parenchyma and the bronchi. The connection between the articles cited in the "Introduction" and the purpose of this study is visible.
The goal is included after the first paragraph of the section. This arrangement and the form of presentation of the goal is the most correct, since the advantages of the presented research system over the existing methods that were described after the goal are immediately visible.
The section "Experimental method" presents methods that allow processing lung images at various stages of the system, including preliminary data processing, lung circulation, pulmonary airway circulation, as well as systematization of data, identification of bronchi and results. All the information of the section is necessary. The design of the study is clear. In this section, the authors make references to previously conducted work, which is necessary to understand the researchers' intention.
In the "Result" section, the main results of the study are presented in separate chapters. All the tasks planned by the authors have been completed. All figures and tables are legible, necessary and complement the content of the section.
In the "Discussion" section, the relevance of the study for lung diseases is presented again. Using the literature, the authors briefly analyzed the known methods of studying light results. In the section, it is necessary to present the system in detail and justify its advantages over other methods. It is necessary to present the limitations of the system, if any.
The conclusion is correct and follows from the results of the conducted research.
The submitted manuscript does not cause any concerns. The manuscript does not cause any ethical problems. All references to publications presented by the authors in the article are necessary and correct, made in the right style. Out of 52 articles, 20 articles for the last 5 years (2017-2022).
I have no concerns about the similarity of this article with other articles published by the same authors.
Author Response
Please see the attachment (Response 2)

Round 2
Reviewer 1 Report
The authors have answered my questions satisfactorily.